# Systems Genome: Coordinated Gene Activity Networks, Recurring Coordination Modules, and Genome Homeostasis in Developing Neurons

**DOI:** 10.3390/ijms25115647

**Published:** 2024-05-22

**Authors:** Siddhartha Dhiman, Namya Manoj, Michal Liput, Amit Sangwan, Justin Diehl, Anna Balcerak, Sneha Sudhakar, Justyna Augustyniak, Josep M. Jornet, Yongho Bae, Ewa K. Stachowiak, Anirban Dutta, Michal K. Stachowiak

**Affiliations:** 1Department of Biomedical Engineering, University at Buffalo, Buffalo, NY 14228, USA; sid.kill3r@gmail.com (S.D.); a.dutta.1@bham.ac.uk (A.D.); 2Department of Pathology and Anatomical Sciences, Jacobs School of Medicine and Biomedical Sciences, University at Buffalo, Buffalo, NY 14228, USA; namyaman@buffalo.edu (N.M.); michal.ubbox@gmail.com (M.L.); jwdiehl@buffalo.edu (J.D.); annabalc@buffalo.edu (A.B.); ssudhaka@buffalo.edu (S.S.); jaugustyniak1@o2.pl (J.A.); yonghoba@buffalo.edu (Y.B.); eks1@buffalo.edu (E.K.S.); 3Mossakowski Medical Research Center, Stem Cell Bioengineering Department, Polish Academy of Sciences, Pawinskiego Str., 02-106 Warsaw, Poland; 4Department of Electrical Engineering, Northeastern University, Boston, MA 02115, USA; amit@northeastern.edu (A.S.); j.jornet@northeastern.edu (J.M.J.); 5Institute of Metabolism and Systems Research, Birmingham Research Park, Birmingham B15 2SQ, UK

**Keywords:** genome integration, entropy, noise and information processing, schizophrenia, development

## Abstract

**Simple Summary:**

A synchronized global genome is a flexible, homeostatic system that underwrites ontogenic development and deprograming in disease.

**Abstract:**

As human progenitor cells differentiate into neurons, the activities of many genes change; these changes are maintained within a narrow range, referred to as genome homeostasis. This process, which alters the synchronization of the entire expressed genome, is distorted in neurodevelopmental diseases such as schizophrenia. The coordinated gene activity networks formed by altering sets of genes comprise recurring coordination modules, governed by the entropy-controlling action of nuclear FGFR1, known to be associated with DNA topology. These modules can be modeled as energy-transferring circuits, revealing that genome homeostasis is maintained by reducing oscillations (noise) in gene activity while allowing gene activity changes to be transmitted across networks; this occurs more readily in neuronal committed cells than in neural progenitors. These findings advance a model of an “entangled” global genome acting as a flexible, coordinated homeostatic system that responds to developmental signals, is governed by nuclear FGFR1, and is reprogrammed in disease.

## 1. Introduction

Systems biology postulates that complex biological systems have emergent properties that cannot be explained solely by the properties of their components [1]. It applies the general systems theory of Bertalanffy [2], and investigates the thermodynamic aspects of living organisms [1]. Cells, organelles, macromolecular complexes, and regulatory and metabolic pathways are examples of biological systems from the micro- to the nanoscopic scale. 

Recently, protein-mediated DNA–DNA interactions between distant chromosome regions, 100 s or 1000 s of kilobases apart and even between DNAs of different chromosomes [3,4] have been revealed through new chromatin conformation capture (3C) and high-throughput interaction assays (Hi-C and HiChIP) [3,4]. The billions of possible interactions among thousands of genes suggest the existence of a coordinate systems genome in which gene activities may be related to one another. 

The decision on cell type fate relies on selective expression of multi-gene programs, as the genome of every cell harbors information for all types of cells in the body. Whereas pluripotent embryonic stem cells develop into all types of cells in the body, tissue-specific multipotent stem cells have restricted potential [5]. For example, multipotent neural stem progenitor cells (NPCs), which reside in the brain ventricles, can produce only brain cells: neurons, astrocytes, and oligodendrocytes [6].

As the populations of human NPCs begin differentiating into neuronal committed cells (NCCs), the activities of a common population of 4646 genes change, including genes in ontogenic programs for “development of the nervous system”, “development of the brain and its parts”, “stem cell self-renewal program”, “cell division and proliferation”, “neuronal differentiation”, “axonal guidance and growth”, “synapse formation”, “neuronal survival”, etc. [7].

Studies over the last three decades have corroborated a pan-ontogenic mechanism known as integrative nuclear FGFR1 signaling engaged in widespread gene programming during cell development [6,7,8,9,10,11,12,13,14,15,16]. The regulatory control is exerted by a nuclear form of FGFR1 protein (nFGFR1) that integrates signals from development-initiating factors and targets thousands of genes encoding mRNAs and microRNAs as well as long noncoding RNAs. nFGFR1 interacts with the common transcription coregulator CBP [8], forms complexes with RAR/RXR, Nurs, and estrogen receptors, and binds to thousands of conserved loci of the mouse and human genome including diverse transcription factor binding elements [6,8,10,13,17,18]. Our recent studies demonstrate the widespread inter- and intrachromosomal interactions differ between embryonic stem cells and NCCs [19]. The differences were apparent in chromatin looping structures and topology-associating domains (TADs), which grouped genes for related ontogenic functions, e.g., proliferation and metabolic functions in embryonic stem cells and neuronal development and transcriptional regulation in NCCs. We proposed a topologically integrated genome archipelago model in which there are extensive transformations through the formation of islands of TADs that comprise genes in changing ontogenic programs [19]. The nFGFR1-targeted genomic sites are concentrated in the borders of TADs and include a sequence that binds chromatin structure-controlling CTCF. nFGFR1 affects the formation of DNA loops and could underwrite/facilitate global genome function via topology-associated domains [19]. 

Studies in several laboratories showed the localization of FGFR1 and other FGFR proteinsin the nuclei of diverse types of cells including fibroblasts [12,20], neurons and endocrine cells [14,21,22], astrocytes [23], developing skeletal cells [24], and different types of cancer [25]. Loss- and gain-of-function experiments [8] have shown that nFGFR1 instructs stem cells to form new neurons and regulates their development in vitro and in vivo. Disruption of nFGFR1 actions in the nervous system has been linked to the neuropsychiatric disorder schizophrenia [10] and disruption in other tissues has been associated with a variety of cancers, including glioma [15,16], pancreatic [17,23,26], osteosarcoma [27], and breast cancer [18,28]. In the present study, we assessed global control of genome function during the developmental transition of NPCs to NCCs, in schizophrenia, the degree to which gene functions are synchronized, and what mechanisms control genome coordination and homeostasis. We examined the effects of nucleus-targeted dominant-negative nFGFR1 and nFGFR1 overexpression [7] to reveal the role of integrative nFGFR1 signaling in global genome function and synchronization.

## 2. Results

### 2.1. Gene Activity during NCC Differentiation: Genome Homeostasis and Role of nFGFR1

We calculated the logarithmic fold changes in average gene expression in three independent biological samples to quantify the differences between the following experimental groups (Methods, Appendix A): group A, 4646 genes whose activity changed when NPCs transition to NCCs; group B, 332 genes whose activity changed by reduction in nFGFR1 function in NPCs [via transfection of a dominant negative nuclear FGFR1(SP-/NLS)(TK-), hereafter referred to as NPC^TK−^]; group C, 861 genes whose activity changed by reduction in nFGFR1 function in NCCs (i.e., NCC^TK−^); group D, 478 genes whose activity changed when NPC^TK−^s transition to NCC^TK−^s (Figure 1A). 

The effects of the dominant negative nFGFR1 on the transitional changes in gene activities were determined by comparing the fold changes for NPC → NCC (group A) with those for NPC^TK−^ → NCC^TK−^ (group D). To determine how gene dysregulation by dominant negative nFGFR1 in NPCs (group C) and/or NCCs (group B) affected these transitional changes, the gene changes common between A∩C∩D, A∩B∩C, and A∩B∩C∩D were determined (Figure 1A). The statistical significance of the transitional fold changes for NPC → NCC and NPC^TK−^ → NCC^TK−^ was determined using a *t* test, and the difference in binned distributions was determined using a Kolmogorov–Smirnov test. 

Figure 1C shows frequency histograms of the log fold changes in gene expression during the transitions of NPCs to NCCs and of NPC^TK−^s to NCC^TK−^s. The frequencies follow a normal Gaussian distribution, with most genes displaying moderate changes and only a few showing extreme changes. However, the distribution of expression changes during the NPC → NCC transition was significantly different from that during the NPC^TK−^ → NCC^TK−^ transition. The genes that differed were those affected by diminished nFGFR1 function in NCCs (A-C-D common group), as neither the A-B-D nor A-B-C-D common group showed a significant effect of reduced nFGFR1 function (Figure 1B and Appendix A). The attenuation of nFGFR1 function specifically in developing NCCs affected gene activities in the NPC^TK−^ → NCC^TK−^ transition by decreasing the number of genes with moderate fold changes and increasing the number of genes with high fold changes. In other words, nFGFR1′s effect increased proportionally to the magnitude of the fold change in gene activity (Appendix A).

The genes common to groups A, C, and D optimally separate into nine clusters, with clusters 3 and 6 representing inhibitory fold changes, clusters 1, 5, 7, 8, and 9 representing moderately activating fold changes, and clusters 2 and 4 representing strongly activating fold changes (Figure 1D). Analyses via *t* tests for both NPC → NCC and NPC^TK−^ → NCC^TK−^ transitions revealed statistically significant differences for genes in clusters 1, 2, 5, 7, 8, and 9. Genes in clusters 1, 2, and 5 displayed low, moderate, and high positive log fold changes, respectively; the expression of most genes increased during the transition when nFGFR1 function was reduced (Appendix A). Notably, the pattern of *RNF170* expression (gene in cluster 1) reversed: the gene was activated (rather than suppressed) during transition with reduced nFGFR1 function (Appendix A). The inhibition of most of the genes in clusters 3 and 6 was stronger in the NCC^TK−^ → NPC^TK−^ transition than in the NCC → NPC transition (Appendix A). The expression of *C7ORF43* (in cluster 3) changed from being suppressed during NPC → NCC transition to enhanced during NPC^TK−^ → NCC^TK−^ transition (Appendix A).

### 2.2. Coordination of Nervous System Development Genes during NPC → NCC Transition: Role of nFGFR1

We assessed the coordinated expression of genes by using RNA-seq data from three independent biological samples. The data were first standardized with a z-score to have a mean of 0 and standard deviation of 1 (example, Appendix A). Coordination was quantified by the Pearson correlation coefficient (*r*). Positive correlations describe gene pairs whose expression activities were both suppressed or enhanced during the transition, whereas negative correlations describe gene pairs whose expression activity changed in opposite directions. Pearson correlation coefficients from standardized RNA-seq data for each experimental group are displayed as frequency histograms in Figure 2, Figure 3, and Appendix A. 

Among the 4646 genes regulated during NPC → NCC transition, 835 genes were in the nervous system development (NSD) GO category. The distributions of NSD gene correlations differed significantly (χ^2^, *p* < 0.0001) between NPCs and NCCs, such that the frequencies of strongly correlated gene pairs (*r* > ±0.96) were higher in NCCs: i.e., more NSD genes were positively or negatively correlated in NCCs (Figure 2A). 

The distribution of NSD gene correlations was significantly affected (χ^2^, *p* < 0.00001) by nFGR1 function, which was observed by comparing NCCs with those that transitioned when nFGFR1 function was reduced (i.e., NCC^TK−^s) (Figure 2B) and with those that transitioned in the presence of constitutively active nFGFR1 [by transfecting NPCs with an FGFR1(SP-/NLS) construct; NCC^NLS^s] (Figure 2C). The frequencies of strongly correlated pairs (negatively and positively correlated) were greater in NCC^TK−^s and lower in NCC^NLS^s, indicating that nFGFR1 activity dampens gene coordination in NCCs. In NPCs, however, the reduction in nFGFR1 function did not significantly affect the correlations (Figure 2D).

The average *r* values (*r*_avg_) (positive or negative) for the pairs of the NSD genes that were strongly coordinated in NPCs were significantly lower in NCCs and in NPC^TK−^s (Figure 2E). Likewise, the *r*_avg_ values for the pairs of NSD genes strongly coordinated in NCCs were much lower in NCC^TK−^s and NCC^NLS^s. Similar results were obtained when examining gene pairs for other ontogenic GO categories (“CNS development” alone or combined with “generation of neurons”) (not shown). Thus, alteration of nFGFR1 activity significantly altered the coordination of gene activity during NPC → NCC transition. 

### 2.3. nFGFR1 Coordinates the Entire Expressed Genome during NPC → NCC Transition

Further analyses showed that the distributions of correlations were significantly different between NPCs and NCCs such that the frequencies of strongly negative or positive correlations were higher for NCC genes. These differences were observed when examining all 16,137 expressed genes (Figure 3A) and when separately examining either the 4646 genes which changed their activities during NPC → NCC transition and were referred to as regulated (Reg) genes (Figure 3B) or the 11,491 genes that did not change their activities and were referred to as not regulated (nonReg) genes (Figure 3C). Furthermore, the cross-correlation of Reg and nonReg genes differed significantly between NPCs and NCCs (Figure 3D). These findings are summarized in Figure 3F.

We also examined how nFGFR1 function affects the coordination of the different gene sets. Reduction in nFGFR1 function significantly altered the distributions of correlations in both NPC^TK−^s (Appendix A–D) and NCC^TK−^s (Appendix A–D) for all 16,137 expressed genes, the 4646 Reg genes, the 11,491 nonReg genes, and the cross-correlated Reg and nonReg gene pairs. Specifically, there were fewer negatively correlated pairs for all gene sets in NPC^TK−^s. In NCC^TK−^s, however, the frequencies of positive correlations increased for the Reg, NSD, and Reg–nonReg genes and decreased for nonReg genes. Overexpression of the active nFGFR1 in NCCs reduced the frequencies of both the strongly positive and negative correlations in all gene sets (except NSD genes) (NCC^NLS^s; Appendix A–D), further indicating that nFGFR1 dampens the coordination of gene activity. However, the role of nFGFR1 in coordinating nonReg genes appears to be different in NPCs (supporting role) than in NCCs (opposing role). The findings from Figure 2, Figure 3, and Appendix A are summarized in Figure 3F.

### 2.4. Global Genome Coordination in NCCs Is Altered in Schizophrenia

Similar to the changes in genome programing during the NPC → NCC transition that we observed, significant changes in the global genome coordination in NCCs was also observed in cells derived from individuals with schizophrenia (Figure 3E). Our earlier studies revealed 1349 dysregulated (Dysreg) genes common among NCCs derived from four patients with schizophrenia [10]. The changes were associated with a loss of nFGFR1 protein in developing cortical neurons in cerebral organoids [7]. The present analyses show significant differences (χ^2^, *p* < 0.00001) in global genome coordination between control and schizophrenia conditions: Dysreg genes in schizophrenia were enriched among strongly positive correlations (Figure 3E_1_). The enrichments were observed also when analyzing all expressed 15,279 genes (Figure 3E_2_), 13,893 nonDysreg genes (Figure 3E_3_), and the cross-correlation of the Dysreg and nonDysreg genes (Figure 3E_4_). These findings are consistent with the global genome coordination model and implicate its relevance in schizophrenia (data summarized in Figure 3F). 

### 2.5. Entropy of Gene Correlation Is Influenced by NPC → NCC Transition, nFGFR1, and Schizophrenia

The significant changes in the frequencies of gene correlations in NCCs compared to that in NPCs were accompanied by changes in the entropy of coordination which are summarized in Appendix A. Whereas the entropy decreased with the correlation changes among all expressed genes, nonReg genes, and in the cross-correlation of the Reg–nonReg genes (indicating that their coordination becomes more ordered in NCCs), the entropy increased for the Reg genes, including NSD genes (Appendix A). The significant differences in the distributions of correlations in NCC^TK−^s (Figure 2 and Appendix A) were associated with increased entropy for the coordination of all expressed genes, Reg genes, and for the cross-correlation of the Reg–nonReg genes but not for the NSD genes (Appendix A). Similarly, increased entropy was noted for all genes in NCC^NLS^s (Appendix A). Thus, any shift in nFGFR1 function makes the genome less organized. (A coordinate gene model inspired by these entropy changes is depicted in Discussion.)

In NCCs from patients with schizophrenia, the coordination for all gene categories (all expressed genes, Dysreg, nonDysreg, and cross-correlated Dysreg–nonDysreg genes) was associated with a decrease in entropy relative to that in NCCs from control individuals (Appendix A), showing a generalized over-synchronization of the genome. 

### 2.6. Reconstruction of the Strongly Correlated Genome during NPC → NCC Transition and in Schizophrenia

To determine what gene activity is coordinated during NPC → NCC transition, we analyzed the strongly correlated gene pairs (those within the 95–100% *r* correlation threshold values). First, the positive (>0 to +1) and negative (<0 to −1) *r* frequency data were fitted to a β model (Appendix A), and the 95% positive and negative confidence threshold values for all expressed genes in different experimental groups were established: values were similar and are listed in Appendix A. Among the gene sets that showed high coordination in NPCs [i.e., all expressed genes, Reg genes, cross-correlated Reg–nonReg genes (Appendix A) and NSD genes (Figure 2E)] the *r*_avg_ values (positive or negative) were significantly lower in NCCs. Likewise, the strongly correlated genes in NCCs had significantly lower *r*_avg_ values in NPCs. Thus, the strongly correlated gene sets in NPCs are replaced by different sets of strongly correlated genes in NCCs. The formations of the strongly correlated gene sets in NPCs and NCCs were governed by nFGFR1, as all strongly correlated expressed genes, Reg genes, cross-correlated Reg–nonReg genes, and NSD genes had significantly lower *r*_avg_ values when nFGFR1 was overexpressed or when nFGFR1 function was reduced (Appendix A). Similarly, different sets of strongly correlated genes occurred in NCCs from patients with schizophrenia and from controls (Appendix A). 

### 2.7. NSD Coordinate GANs Are Reconstructed during NPC → NCC Transition: Role of nFGFR1

To determine if the significant changes in NSD *r*_avg_ values (Appendix A) reflect different coordinate GANs, we exported the correlation pairs with *r* values of >0.99 to Cytoscape [29], which displays gene names (nodes) in a circle; lines connect those with strong positive correlations (Materials and Methods). 

The GAN in Figure 4A shows positive correlations among 83 Reg genes for which activity changed during NPC → NCC transition (results summarized in Figure 4D). These genes formed a smaller GAN in NCCs, with 9-fold fewer connections (Figure 4A, middle) and a decrease of 59% in the clustering coefficient. Although all 83 genes were included in the GAN for NPC^TK−^s, there was a 47% decrease in the number of neighbors compared to that in NCCs and the clustering coefficient was 19% lower (Figure 4A, right; Figure 4D). Thus, the reduction in nFGFR1 function disrupts the coordination of NSD gene activity.

The 131 most highly coordinated (*r* > 0.99) NSD genes in NCCs, whose expression was influenced by nFGFR1 function, formed two GANs linked by *PCDH15* and *SEMA3D* (Figure 4B,D). In NPCs, the same 131 genes had >9-fold fewer connections and average neighbors and a 27% lower clustering coefficient (Figure 4D). Thus, there were separate GANs for NPCs and NCCs. More than 50% of the genes were excluded from the GAN for NCC^TK−^s, with 78% fewer average neighbors and 27% lower clustering coefficient (Figure 3D), while the remaining genes formed associations with only with a few other genes and smaller GANs (Figure 4B). Thus, highly coordinated GANs in NPCs and NCCs were deconstructed when nFGFR1 function was reduced, while new GANs were formed (Appendix A). 

Ninety of the NSD genes whose activity was affected by constitutively active nFGFR1 function formed a single GAN in NCCs (Figure 4C,D). In NCC^NLS^s, this GAN had 33% fewer connections, 25% fewer node neighbors, and an 11% lower clustering coefficient. On the other hand, the overall GAN that formed in NCC^NLS^s had ~33% fewer connection and node neighbors than the GAN in control NCCs (Appendix A). Altogether, the data show that any shift in nFGFR1 signaling deconstructs NSD GANs and promotes the formation of the new GANs.

### 2.8. Intra- and Intercluster Synchronization of Functionally Related GAN Genes

To investigate the biological consequence of gene coordination in the identified NSD GANs, we first performed unsupervised hierarchical clustering to highlight the relationships among specific genes. The gene functions (listed in Appendix A) were assigned based on the analyses using g:Profiler sapiens and Reactome and are listed in Appendix A. Figure 5A (top) shows that the 83 genes (listed in Appendix A) of the NPC GAN (see Figure 4A) formed ten color-coded clusters, with five non-clustered connector (Ct) genes; genes in this GAN are involved with mitochondrion-related apoptosis, mitotic cycle, cell adhesion, extracellular matrix remodeling, and synapse formation (Appendix A). Several pathways were associated with two or more clusters, such as pathways for nerve growth factor signaling, tropomyosin-related kinase A, and mitogen-activated protein kinase signaling (clusters I, III, and VI), clathrin-mediated endocytosis (clusters III, VIII, and X), Wnt signaling (clusters VII, VIII, X, and connectors), Notch signaling known to prevent neuronal differentiation (clusters II and X), and survival (three anti-apoptotic humanin genes were in cluster I, two in cluster III, and one in cluster X). This organization indicates that the NPC GAN has genes with complementing developmental functions via intra- and intercluster gene synchronization. Many of the genes that formed the clusters in NPCs were organized in a markedly different, less-coordinated (less-connected) manner in NCCs, and 45 of the genes were outside the network, including genes 1, 2, and 4–8 (listed in Appendix A), which had the most connections (52) in NPCs (Figure 5A, middle). 

The NPC network and clusters changed markedly when nFGFR1 function was reduced (Figure 5A, bottom). For instance, genes from NPC clusters II, III, and IV became dispersed throughout the NPC^TK−^ network, indicating that nFGFR1 coordinates the expression of genes important for cell adhesion, axon guidance, synaptic formation, mitochondrion-related apoptosis, and Notch and neurotrophic tropomyosin-receptor kinase signaling. 

We then analyzed the GANs of the strongest positively correlated (*r* > 0.99) NCC genes affected by reduced nFGFR1 function (131 genes) (Figure 5B, bottom; Appendix A); 20 of the genes had significantly fewer connections (40.7 ± 1.52; *p* < 0.001) than in the NPC GAN (Figure 5B, top; Appendix A). Additionally, 20 of the 90 genes that were strongly positively correlated in NCCs and affected by nFGFR1 overexpression (Figure 5C, bottom; Appendix A) had significantly fewer connections (34.93 ± 5.28; *p* < 0.0001) than in the NPC GAN (51.05 ± 1.36, Figure 5B, middle; Appendix A). Thus, in general, the GAN was more complex in NPCs than in NCCs.

The 131-gene NCC GAN consisted of 12 clusters plus eight non-clustered connector genes (Figure 5B, top; Appendix A). Four genes encoding Wnt proteins and two Wnt signaling genes were concentrated in clusters VI and XII, with the additional Wnt signaling genes in clusters IV, V, and VIII. Nerve growth factor/neurotrophic tropomyosin-receptor kinase signaling genes were associated with clusters I, III, and VI, and axon guidance genes were associated with five clusters: one in cluster I, two in clusters II and X, four in cluster III, and one was a connector gene. Multiple transcription regulators were represented in clusters II–VI, VIII, and X–XII and two were connectors. Of the 131 NCC connected genes, 92 were excluded from the GAN in NPCs; the remaining genes were dispersed throughout the network (Figure 5B, middle). Thus, the NCC clusters form during NPC → NCC transition.

Reduction in nFGFR1 function fractured the NCC GAN into seven disconnected networks, a single two-gene pair, and 13 disconnected genes (Figure 5B, bottom). The latter included transcription factor genes *PAX7* (node #8), *MYC* (#114), and *LEF1* (#128), *SLFN13* for RNA processing (#70), lncRNA DIO30S (#97) known to be associated with glioma, *PCDH15* for a protocadherin (#120), *LRP2* encoding a lipoprotein-receptor-related protein (#66), *RTN1* for a reticulon (#123), *PRKCH* encoding protein kinase C (#54), *PLG2* for a plasminogen (#62), and *USH1G* for a scaffolding protein (#85). The remnants of the highly coordinated NCC clusters I–III either separated from the main cluster (note the spread of red, green, and yellow nodes) or remained in the group in reduced numbers. Hence, reducing nFGFR1 function in NCCs disrupts a highly synchronized clustered GAN. 

The NCC network formed by the 90 genes affected by nFGFR1 overexpression formed four clusters with 14 connector genes (Figure 5C, top; Appendix A). A striking feature in this network is the grouping of 17 genes for transcription factors in a single cluster (cluster IV). In NCC^NLS^s, small networks of two to four connected nodes formed outside the main network (Figure 5C, bottom), and 14 genes were disconnected from the network: from cluster I, *SYT13* for synaptotagmin (node #72), *NR4A2* for a transcription factor (#74), and *S100B* for nuclear signaling (#85); from cluster II, *PRKCH* for G-protein coupled receptor signaling (#59), and genes encoding protocadherin for axon guidance-cell contact (#60) and synaptotagmin (#90) involved in synapse formation; cluster III, *SNPH* for control of mitochondrial function (#47), *TNFRSF21* for a tumor necrosis factor receptor for control of apoptosis (#53), a gene for synaptic neuromedin (#71), *BARLH2* encoding a transcription factor (#78), and *EID1* for a global transcription coregulator (#84). Genes from cluster IV were dispersed throughout the NCC^NLS^ network, connected by longer edges (less coordinated), and forming fewer edges than in control NCCs: two genes from cluster IV encoding transcription factors, *FOS* (#10) and *NR4A3* (#15), were no longer connected with other network genes. Hence, excessive nFGFR1 signaling alters the formation of a synchronous network of genes of complementary neurodevelopmental functions.

We also analyzed changes in the strengths of the positive interactions between selected gene pairs critical for neural development by comparing their correlation coefficient values in the experimental groups. In general, the interactions observed in NPCs were absent in NCCs, and vice versa. Furthermore, many of the positive correlations were eliminated or altered by shifts in nFGFR1 function (Appendix A–C).

### 2.9. Cross-Coordination of Transcription Factor Genes and Neurodevelopmental Genes Is Changed during NPC → NCC Transition and Influenced by nFGFR1 

In the feed-forward-and-gate model [6], nFGFR1 controls neurodevelopmental genes directly by targeting their promoters as well as indirectly by targeting and regulating promoters of the upstream TF transcription factor genes. Hence, we next examined if nFGFR1 controls cross-correlation between diverse transcription factor genes and other neurodevelopment genes. We focused on a subgroup of the 4646 Reg genes (whose activity changed during NPC → NCC transition) that had the strongest correlation (*r* > ±0.96) between neurodevelopmental and transcription factor genes (Figure 6). In NCCs, the numbers of these strongly correlated gene pairs increased 2.5-fold for those with *r* values > +0.96 and 1.7-fold for those with *r* values < −0.96; the *r*_avg_ (−0.145) shifted significantly to a positive value (+0.0816). Both the numbers of pairs and *r*_avg_s were reduced in NCC^TK−^s and in NCC^NLS^s (Figure 6B). Entropy of the cross-correlation matrices (Figure 6 and Appendix A) changed positively in NPC^TK−^s versus NPCs, in NCC^TK−^s, and in NCC^NLS^s versus NCCs. Thus, nFGFR1 promotes synchronization of neurodevelopmental and transcription factor genes. 

Figure 6A shows a heat map of the strongly correlated transcription factor genes and neurodevelopmental genes in NCCs. Genes for transcription factors such as CREB and the families of EGR, Fos, Fox, GLIS, HES, Jun, Sox or TEADs, and ZNF display similar patterns of coordination (positive or negative) similar to those for with the neurodevelopmental genes; these are referred to as group I. The pattern of cross-correlations formed by transcription factors of the ZNF family, ZBTB, and ZFP (referred to as group II) were mostly opposite to that of group I. These distinct patterns were largely absent in NPCs, which suggests that transcription factors in groups I and II act in concert to program the neurodevelopmental genes during NPC → NCC transition. However, the functions of groups I and II appear to be opposite. The mirror-like patterns of the NCC gene cross-coordination were disrupted in NCC^TK−^s and in NCC^NLS^s (Figure 6A). 

In NPCs, genes for a diversity of transcription factors cross-coordinated with those encoding humanin-like neuroprotective mitochondrial proteins located on different chromosomes: *MTRNR2L4* (Ch 16), neuroprotective *MTRN2RL8* (Ch 11), and *MTRN2RL6* (Ch 7). This pattern was disrupted in NPC^TK−^s (Figure 6A).

An analysis of transcription factor and neurodevelopmental genes of the entire population of 4646 Reg genes (Appendix A) verified that the numbers of positively or negatively cross-correlated gene pairs doubled, their *r*_avg_ values were significantly higher in NCCs than in NPCs, and both were reduced with diminished nFGFR1 function. The average negative cross-correlation in NCCs was also reduced by nFGFR1 overexpression. 

We conclude that the synchronized cross-coordination of the transcription factor genes with their effector neurodevelopmental genes depends on the developmental stage and is dictated by nFGFR1 (i.e., affected by diminished nFGFR1 function in NPCs and NCCs and by overexpression of nFGFR1 in NCCs). Thus, any shift in nFGFR1 signaling leads to the formation of less-organized (higher entropy) transcription factor neurodevelopmental gene systems.

### 2.10. GANs Comprise Recurring Coordination Modules (RCMs): The Distribution of RCMs Changes during NPC → NCC Transition and Is Programmed by nFGFR1

The directional flow of information through the cascades of transcription factor genes in gene regulatory networks proceeds through a relatively small set of recurring simple regulator motifs [30], which are organized into dense overlapping regulons to underwrite genome regulatory function. These recurring simple regulator motifs were identified with open-source software developed by Kashtan et al. [31]. To determine whether nondirectional gene coordination in GANs also involves recurring coordination modules, we used FANMOD software (fanmod-windows.zip) [32], which detects motifs as patterns that occur more frequently in a given network than in random networks of the same size and with the same connectivity properties. The software accepts network data (list of the interactions between different genes) and outputs the recurring motifs within the network. Given the limitations of the available computational power, we investigated motifs composed of three to six gene nodes. The significantly overrepresented RCMs are listed in Appendix A along with their entropy values.

To analyze the distribution of the RCMs in NPC and NCC GANs, we binned the RCMs on the basis of their complexity (number of edges). The frequency data were converted to a logarithmic scale to analyze patterns among the groups. The best trend line for six nodes was graphed (Figure 7A,B); separate graphs for three to six nodes are available in Appendix A. A Shapiro–Wilk test showed that the RCM frequency curves in NCC and NPC GANs were not normally distributed; therefore, a nonparametric Wilcoxon rank sum test was applied and showed that (i) the RCM distribution in the NPC GAN (6-node RCM, Figure 7A; and 3- to 6-node RCMs, Appendix A) was significantly different from the distribution in the NCC or NPC^TK−^ GAN, and (ii) the RCM distribution within the NCC GAN (6-node RCM, Figure 7B; and 3- to 6-node RCMs, Appendix A) was significantly different from the distribution in the NPC or NCC^TK−^ GAN.

The 83-gene NSD GAN of NCCs B and Figure 4B) formed six-node RCMs of low complexity (3–6 edges) with intermediate entropy (S_avg_ = 0.893), RCMs of medium complexity (7–12 edges) with high entropy (S_avg_ = 0.9729), and RCMs of high complexity (13–15 edges) with low entropy (S_avg_ = 0.7555) (Figure 6A). The frequency of RCMs in NPCs declined between the low- and high-complexity RCMs (from the simple 5-edge RCM to the most complex 15-edge RCM). After the transition to NCCs, the medium-complexity/high-entropy RCMs became less frequent, and the formation of medium-complexity/high-entropy and high-complexity/low-entropy RCMs was reduced in NPC^TK−^s. Thus, in the NPC GAN, nFGFR1 promotes the formation of medium-complexity/high-entropy and high-complexity/low-entropy RCMs.

The 131-gene GAN of NCCs (Figure 3B and Figure 4B) formed six-node RCMs of medium complexity (7–12 edges) with high entropy (S_avg_ = 0.9789) and RCMs of high complexity (13–15 edges) with low entropy (S_avg_ = 0.7798). However, unlike the NPC GAN, the NCC GAN did not contain low-complexity (3–6 edges)/intermediate-entropy RCMs. There were fewer occurrences of the NCC medium-complexity/high-entropy RCMs and the high-complexity/low-entropy RCMs in NPCs, indicating that these NCC RCMs form largely during the NPC → NCC transition. Frequencies of the medium-complexity/high-entropy NCC RCMs were reduced markedly in NCC^TK−^s. Thus, in the NCC GAN, nFGFR1 promotes the formation of medium-complexity/high-entropy RCMs. 

The patterns noted above were reproduced for three-node to six-node RCMs analyzed together (Appendix A), further demonstrating the role of nFGFR1 in the formation of medium-complexity/high-entropy and high-complexity/low-entropy RCMs in NPCs and medium-complexity/high-entropy RCMs in NCCs. 

### 2.11. Modeling RCMs as Information-Processing Circuits: RCMs Influence Regulatory Noise and Signal Transmission

The dynamics of gene activity responses are an inherent source of noise during activity oscillations [33]. One potential role of the RCMs is to mitigate this problem. To examine this, we modeled the signal (information)-processing functions of RCMs as an equivalent electrical R-L-C circuit, with an individual gene-node acting as an inductor (L), resistor (R), and capacitor (C) (Figure 7C). Alternatively, the functions could be modeled as an equivalent mechanical system (see Appendix A). In an electrical circuit, R causes a loss of supplied energy (and restricts the flow); in the genome function circuit, R reflects the innate resistance of gene promoters and epigenomic modifications. In an electrical circuit, L is a phasor element that initially blocks alternating current before sending it through with an added phase/delay; in the genome function circuit, L reflects the time required to evoke the change in activity of the next node (gene) in the network, causing a delay and a change of phase. In the electrical circuit, C is also a phasor element and stores energy; in the genome function circuit, C reflects mechanical energy stored in looped or supercoiled DNA as well as subthreshold accumulation of an effector. In an electrical circuit, oscillations are produced by L and C, which also decide the frequency and sensitivity to the oscillations. Using transfer function (see Appendix A, Computational Methods, *Modeling RCMs as information-processing resistor (R), inductor (L), capacitor (C) circuit)* we can calculate the transfer function of the network by cascading them as follows: $$H\left(s_{1},s_{2}...\right)=H\left(s_{1}\right)H\left(s_{2}\right)...$$

Figure 7D and Appendix A illustrate the step responses of different complexity R-L-C circuits (modeled on detected RCMs) to a transient stimulus according to R-L-C network theory (see Appendix A). The R-L-C model predicts that 14-node (edge) RCMs will have lower effective L than 6-edge RCMs. More densely connected nodes (such as 10- and 14-edge inductor networks in R-L-C circuits) have reduced L that decreases oscillations, effectively increasing the overall dampening in the circuit. In a gene network, a more interconnected gene could thus provide resilience to stimuli fluctuations and regulate genomic functions by absorbing some of the energy from oscillations. Thus, highly connected RCM genes may dampen the excess energy oscillations and thereby prevent corruption of the information in GANs.

We next considered the transmission of signal (information) between the RCMs. For simplicity, we considered transmission between identical RCMs (for detail, see Appendix A). Figure 7E shows the dampening function of 6, 10, and 14 edges for a two-level RCM system and the effect of such dampening on the flow of information into the next module. In a low-dampened network (six edges), low “subcritical” levels of intramodular connectivity are associated with excessive network noise, and although the information is transmitted, it may be corrupted by node oscillations. The RCM with the highest “overcritical” connectivity (16 edges) would dampen the noise but also diminish or eliminate the transmitted signal, thereby blocking the flow of information. An intermediate “critical” state of intramodular connectivity (10 edges) reduces network noise while allowing the transmission of uncorrupted signal within the GAN. 

The model illustrates the potential effects of intramodular connectivity on the communications between the RCMs and, thus, the overall flow of information through the GANs. As can be inferred from the graphs in Figure 7A,B and Appendix A, nFGFR1 promotes the critical and overcritical connectivity in NPCs and critical connectivity in NCCs. The influences of low dampening, critical dampening, and overcritical dampening on information transfer are further illustrated in Appendix A (video).

## 3. Discussion 

The present study brings to the forefront the concept of the systems genome, in which gene activities (i.e., during cell development) are coordinated under global control. This control promotes moderate changes and counteracts extreme changes in gene activities. This process is actively regulated by nFGFR1, which acts as a “band-pass filter” to maintain genome activity within a set range (i.e., genome homeostasis). 

Although studies of development tend to focus on genes with the greatest change in activity, many more genes are subject to moderate up- or downregulation, and the majority of genes do not change significantly their activity. These so-called “background bystanders” were not considered active participants in genome regulation. The present study challenges this view by revealing widespread coordination among expressed genes that is subject to developmental regulation.

### 3.1. Entropy-Based Genome Synchronization Model: Over-Synchronization in Schizophrenia

Genome homeostasis was associated with an enrichment of highly correlated (positively and negatively) genes within the population of 16,167 genes expressed in NPCs and NCCs; gene sets included 4646 Reg genes (whose expression was altered during NPC → NCC transition), a subset comprising NSD genes, 11,136 nonReg genes (whose activity was not altered during transition), and a subset of Reg–nonReg cross-correlated genes. In all these gene populations, the distribution of gene coordination was significantly altered between NPCs and NCCs, with an additional enrichment of highly correlated gene pairs in NCCs and different sets of NPC and NCC genes forming the highly coordinated pairs. 

Alterations in the frequencies of gene correlations (Figure 2A–D, Figure 3A–E, and Appendix A) during NPC → NCC transition were associated with changes in overall correlation entropy (summarized in Appendix A). These findings inspired an entropy-based genome synchronization model (Figure 8A) in which the activity of nonReg genes becomes more synchronized (lowering their entropy) during the transition, while the activity of Reg genes becomes less synchronized (increasing their entropy) but increasingly more cross-synchronized with the activity of nonReg genes (lowering entropy). This process is driven by endogenous nFGFR1 and further augmented by an excess of nFGFR1. The nFGFR1-dependent cross-coordination and cross-synchronization of the Reg–nonReg genes could potentially underwrite the nFGFR1 band-pass filter-like control of genome function during NPC → NCC transition. In general, with the lower entropy of Reg–nonReg gene cross-synchronization, nFGFR1 ensures more useful energy is available for other biological processes, such as global gene regulation. By the same token, the increased entropy of Reg gene synchronization in NCCs indicates that nFGFR1 raises their regulatory flexibility.

Changes in global genome coordination in schizophrenia affect neuron and brain development, which can be modeled in NCCs and cerebral organoids derived from patient cells (induced pluripotent stem cells) [10]. nFGFR1 is dysregulated in NCCs from patients with schizophrenia [7,10], which we observed in the present study as radically increased gene coordination associated with the entropy-based changes, marking a global genome over-synchronization (Figure 3E). The over-synchronization was observed among the Dysreg and the nonDysreg genes and as an increased cross-synchronization of Dysreg and nonDysreg genes (Figure 3E and Appendix A). Thus, in schizophrenia, genome dysfunction in developing neurons generates a differently synchronized (generally over-synchronized) global genome. The role of nFGFR1 in this process needs to be further investigated. The augmented synchrony between Dysreg and nonDysreg genes raises the “egg or hen” question about which is the primary cause of genome dysregulation and etiology of schizophrenia (Figure 8A). An additional question raised is whether the global dysregulation paradigm applies to other diseases, including many types of cancer in which nFGFR1 is dysregulated [17,18,23,27] and thus could be targeted for new therapies. 

In the systems genome proposed here, changes in gene synchronization during development or in disease extend beyond the regulated genes to engage the entire genome. Such an “entangled” global genome could act as a flexible coordinated system that responds to developmental signals and is reprogrammed in disease. However, new approaches to control the coordination of large populations of genes are needed to determine the significance of these global changes in genome coordination. Perhaps targeting recently identified genomic DNA-interacting sequences [19] and controlling chromatin interactions in defined nuclear/chromatin loci (such as with novel optogenomic tools [10]) will bring us closer to this challenging goal. 

### 3.2. GANs

Highly coordinated genes were also overrepresented within the ontological Reg gene subcategories, including NSD genes, which form different highly coordinated pairs in NPCs and NCCs. For each set of genes, the circular layout in Cytoscape [29] facilitated an intuitive representation of the connectivity and co-regulation among highly correlated genes, underlining the influence of nFGFR1 functional changes on gene expression networks. This methodological approach provided a robust framework for comparing gene activity under various experimental manipulations, illustrating the pivotal role of nFGFR1 in regulating gene networks across different cellular contexts. We refer to these as GANs to differentiate them from the directional gene regulatory networks [30] that serve to transmit information (stimulation or inhibition) from the first “receptive” genes to the final “effector” genes. 

Deconstruction of NPC GANs and construction of NCC GANs during transition were represented in their circular networks, in which gene activity coordination was marked by node connections. The deconstruction of NPC GANs was represented by reduced numbers of gene neighbors and lower clustering coefficients (Figure 3A). They were replaced, however, by highly correlated GANs in NCCs, with more gene neighbors and higher clustering coefficients (Figure 4B). The hierarchical networks show that GANs contain clusters of functionally related genes representing different developmental pathways and subject to intra- and inter-cluster coordination. The NPC clusters include genes for mitochondrion-related apoptosis, mitotic cycle, cell adhesion, extracellular matrix remodeling, and synapse formation. NGF, TRK, and MAPK signaling genes were highly coordinated in both NPCs and NCCs. An interesting feature of the NCC GANs is the inclusion of genes for the Wnt/FZD pathway, for transcription factors PAZ7 or MYCL1 centered around control of neural development, and for TRM71, which regulates mitochondrial RNA and microRNA biogenesis. Moreover, highly coordinated clusters comprise multiple transcription factor genes, indicating that their coordinated actions are important for cell differentiation. 

Shifts in nFGFR1 function in NPCs and NCCs were associated with extensive disruption or remodeling of the highly synchronized clustered GANs, indicating that nFGFR1 controls the formation of hierarchical gene networks. nFGFR1 coordinates the activities of transcription factor genes with those of other developmental genes (Figure 5 and Appendix A). For example, two groups of transcription factors in NCCs had miror-like patterns of correlations with neurodevelopmental genes (group I and group II), indicating they exert antagonistic positive and negative control of the neurodevelopmental genes (Figure 5). These correlation patterns are set by the endogenous levels of nFGFR1, as they were deconstructed by either diminished function or overexpression of nFGFR1. We conclude that nFGFR1, which targets the promoters of many of the GAN genes, underwrites the coordinate transcriptome by shifting the balance of positive and negative cross-coordination and lowering the corresponding entropy.

In our investigation of NPC → NCC differentiation, we have focused primarily on the *bona fide* neurodevelopmental pathways. Neurons produce and respond to a variety of cytokines and other pro-inflammatory molecules which control their development [34] and may play a role in schizophrenia [35]. The nFGFR1-controlled GANs include genes which encode cytokine receptor TNFRSF21, signaling molecule ANKRD18DP, MAPK proteins, Notch, WNT, and WNT receptors, all known to mediate actions of neuroinflammatory proteins. Furthermore, in cerebral organoids, the iPSCs develop also to microglia, astrocytes, and oligodendrocytes, all of which are controlled by FGFR [36,37,38] and may contribute to the cortical maldevelopment in schizophrenia. For example, activation of microglia by TNF leads to cortical malformations similar to those observed in the schizophrenia organoids [38].

### 3.3. RCMs Are Kernels of GANs

In gene regulatory networks, simple integrative motifs (i.e., feedback motifs, feed-forward-and-gate motifs, etc.) can combine into dense overlapping regulons that transduce and often sustain irreversible developmental decisions; these combined motifs may persist after the input signal has vanished [30]. Analogous to these integrative motifs, RCMs of GANs may combine into larger genome groupings, which we call “coordinons.” We modeled RCMs as R-C-L circuits, which enabled us to show that an increasing complexity of RCMs progressively eliminates gene activity fluctuations and dampens activity noise to ensure genome homeostasis. In the proposed model, an overly complex RCM acts as a genome low entropy sink that extinguishes the incoming information flow, whereas a less complex (critical complexity) higher entropy RCM allows the information to reliably continue on while minimizing noise and thus disinformation (i.e., gene activity).

In NPCs, the RCM genomic sink promoted by nFGFR1 could maintain a nondifferentiated state by resisting stimuli that extinguish cell renewal networks and initiate differentiation. On the other hand, the critical complexity RCMs in NCCs promoted by nFGFR1 facilitate transmission of developmental signals across GANs to allow neuronal differentiation while minimizing disinformation noise. Similar models have been proposed for brain activity networks in which an overly complex network with low entropy is less amenable to functional modification than a less complex, higher entropy network [39]. 

### 3.4. nFGFR1 Coordination of Gene Activities May Reflect Control of Genome Topology

The sprawling number of interactions among thousands of genes and non-genic genome regions implies a vastly connected systems genome in which even distant groups of genes can be physically connected and their activities coordinated [3,4,19]. The growing size of the genome throughout evolution was likely accompanied by mechanisms to coordinate the expression of genes at different loci, thereby counteracting dyscoordination and a rise in entropy. One such mechanism has been linked to integrated nFGFR1 signaling, in which the coordination of diverse signals enables concerted changes in gene activities [6,8,13]. FGFR1 has an atypical transmembrane domain that allows the newly translated receptor to be released from the pre-Golgi membrane and translocated into the nucleus along with its NLS-containing ligands where it associates with RNA co-transcriptional processing of nuclear speckles [8] and targets thousands of chromatin sites [10,17,18,28,40]. Indeed, nFGFR1 was recently shown to be enriched at topology-associated domains and DNA loops and influence their formation along with CTCF [19]. Moreover, nFGFR1 binds and activates a common transcriptional cofactor (CBP), thereby triggering diverse signaling paths for a concerted global genome response [8].

### 3.5. Is nFGFR1 a Proportional-Integral-Derivative Controller?

By receiving feedback from a variety of receptors and signaling pathways, nFGFR1 could maintain homeostasis and modulate the genome responses akin to a proportional-integral-derivative controller. In general, the proportional-integral-derivative controller maintains a system setpoint by registering a system’s output and taking action to minimize any departure from the setpoint. In our proposed model (Figure 8B), nFGFR1 maintains genome responses at a specific setpoint: nFGFR1 registers the genome output through the feedback signals that control integrative nFGFR1 signaling and coordinates GANs (gene–gene coordination and fold changes) to match the evolutionary set point, thus maintaining homeostasis of the global systems genome. Although nuclear translocation and functions of FGFR1 are controlled by a variety of developmental signals (influenced by genome activities), it is not known whether the actions of FGFR1 are proportional to the difference between the setpoint and registered state.

In summary, the present investigation advances a systems genome paradigm in which genes with small fold changes underscore the ontogeny, and their coordination with the rest of the genome (including the nonReg genes) governs the genome’s responses. The present study also advances the role of nFGFR1 as a global gene activity coordinator and a band-pass filter that maintains gene activities and their developmental regulation within a set homeostatic range. nFGFR1 does this by controlling the information noise and transfers via the formation of RCMs and GANs.

## 4. Materials and Methods

### 4.1. Experimental Design

The RNA-seq databases generated in our earlier studies [7,10] were used in the analyses of the present study. These experiments are summarized below.

### 4.2. Cell Cultures and Treatments

The analyses were performed using RNA-seq data sets [7] from homogenous cultures of NPCs and NCCs differentiated from H9 human embryonic stem cells. To induce NCCs, NPCs were treated with 20 ng/mL BDNF (Peprotech), 20 ng/mL GDNF (Peprotech), 1 mM dibutyryl-cyclic AMP (Sigma), and 200 nM ascorbic acid (Sigma) [10] for 2 days. To study the role of nFGFR1, recombinant DNA constructs encoding FGFR1 were used as previously described in the earlier reports. The FGFR1 (SP-/NLS) construct was used in which the signal peptide (SP) is replaced with a nuclear localization signal (NLS), for constitutive active nuclear nFGFR1 that does not require NLS containing ligand for the nuclear translocation [11,12]. To reduce nuclear nFGFR1 function, a dominant negative variant of the nuclear receptor, FGFR1 (SP-/NLS) (TK-), was used, generated by deleting the C-terminal tyrosine kinase domain (TK-); the construct competes with endogenous nFGFR1 for binding with CBP and DNA targets, thus preventing nFGFR1-regulated transcription [11].

NPCs were transfected with a control β-galactosidase-expressing plasmid, dominant negative FGFR1 (SP-/NLS) (TK-) (abbreviated as TK-), or constitutively active FGFR1 (SP-/NLS) (abbreviated as NLS). After 24 h, cells referred to as NPCs were incubated for 48 h in non-differentiating medium, whereas the cells referred to as NCCs were incubated for 48 h in neuronal differentiation medium with cAMP, BDNF, and GDNF. 

Appendix A provides a summary of the five conditions: NPC, NPC^TK−^, NCC, NCC^TK−^, and NCC^NLS^. Experiments were performed using three biological samples from separate cell culture and transfection experiments, and the average gene expression was calculated from three independent biological samples [7]. RNAs of 4636 genes showed significant differences in expression between NCCs and NPCs (and thus are referred to as regulated [Reg] genes). Within this group of Reg genes, 332 genes showed significant difference in expression between NPCs and NPC^TK−^s, 861 genes showed significant difference in expression between NCCs and NCC^TK−^s, and 440 genes showed significant difference in expression between NCCs and NCC^NLS^ s. These results were included in our previous report [7], and the RNA-seq data were deposited in NCBI GEO with accession code GSE103307. 

To analyze changes in gene expression in differentiating NCCs under conditions of schizophrenia, we used RNA-seq data from the NCCs developed from induced pluripotent stem cells from four patients with schizophrenia having different genetic abnormalities and from four control individuals (GSE92874) [10]; 1349 RNAs from all 15,279 expressed genes showed a significant difference in expression (and thus are referred to as schizophrenia dysregulated [Dysreg] genes) [10].

In both data sets, changes in gene expression were quantified as logarithmic fold change. In both studies, the significant differences were identified as having a false-discovery rate *q* value of <0.05 [7,10]. 

### 4.3. Analysis of the Coordination of Gene Activities

We assessed the coordinated expression of genes by using RNA-seq data from three independent biological samples. The data were first standardized with a z-score to have a mean of 0 and standard deviation of 1 (example, Appendix A). Coordination was quantified by the Pearson correlation coefficient (r) ranging from −1 to +1, thus providing both magnitude and direction of the coordinated expression. Positive correlations describe gene pairs whose expression activities were both suppressed or enhanced during the transition, whereas negative correlations describe gene pairs whose expression activity changed in opposite directions. The differences between the analyzed groups were evaluated with χ2 tests. 

### 4.4. Computational Methods and Statistical Analysis

Computational methods are detailed in Appendix A and in Section 4.5 Statistical analyses were performed as referred in the text, and included *t* tests, ANOVAs, χ^2^ tests, the Kolmogorov–Smirnov test of normality, and nonparametric Wilcoxon rank sum tests.

### 4.5. Complex Network Analyses

Cytoscape is an open-source platform designed to visualize complex networks and integrate them with various types of attribute data [29]. We used Cytoscape to calculate the complex network characteristics including the clustering coefficient, which is the degree to which nodes tend to cluster. The circular graph layout in Cytoscape aims to enhance the visualization of group and tree structures within a network. This layout algorithm employs a method of partitioning the network based on connectivity data, organizing these partitions into distinct circles. In Cytoscape, we assessed the connectivity structure of the network—nodes and their links (edges)—to form clusters that are then visually separated into circles. This clustering is typically based on connectivity metrics, such as node degree or betweenness centrality, which determine how nodes are grouped together. The hierarchical layout is designed to depict the main direction or flow within a network, which is useful for illustrating data with inherent directional or hierarchical relationships (e.g., gene regulatory networks). Nodes are placed on different layers, with the goal of minimizing the number of crossings between edges that connect these nodes. This involves sorting nodes within each layer in a way that aligns outgoing edges downward to the next layer with minimal crossing which includes layer assignment, crossing reduction, and coordinate assignment steps.

We used the following circular network characteristics from Cytoscape [29] (see Appendix A for further details).

**Number of Nodes and Edges:** These are the fundamental units of a network, representing entities and their connections, respectively.**Average Number of Neighbors:** This metric calculates the average connectivity of nodes, reflecting the typical structural environment of a node within the network.**Network Diameter:** Defined as the longest shortest path between any two nodes in the network, this metric indicates the “largeness” of the network’s scope.**Shortest Path Length (Characteristic Path Length):** This is a key metric in network theory, indicating the minimum path length between two nodes, and is crucial for understanding the efficiency of network connectivity. The average of these shortest paths across all pairs is the characteristic path length.**Clustering Coefficient:** This metric describes the likelihood that nodes adjacent to any given node are also connected to each other, forming a cluster.

The clustering coefficient for a node n in an undirected network is mathematically expressed as: $$C_{n}=\frac{2e_{n}}{k_{n}\left(k_{n}-1\right)}$$

Here, $k_{n}$ represents the number of neighbors of n, and $e_{n}$ is the number of edges that exist between all pairs of neighbors of n.

In directed networks, the formula for the clustering coefficient adjusts slightly: $$C_{n}=\frac{e_{n}}{k_{n}\left(k_{n}-1\right)}$$

For both types of networks, the clustering coefficient, $C_{n}$, is essentially a ratio, $\frac{N}{M}$, where: -N is the actual number of edges among the neighbors of n;-M is the total possible number of edges that could exist among the neighbors. 

This coefficient is a value that ranges from 0 to 1, representing the extent to which nodes in a network cluster together. A value of 0 indicates no clustering, while a value of 1 denotes maximum clustering.

**Network Density:** This normalized metric reflects the ratio of actual connections to possible connections in the network, providing insight into how densely the network is populated with edges.**Connected Components:** A measure of the network’s overall connectivity, indicating how many sub-networks exist that are not interconnected.

## Figures and Tables

**Figure 1 ijms-25-05647-f001:**
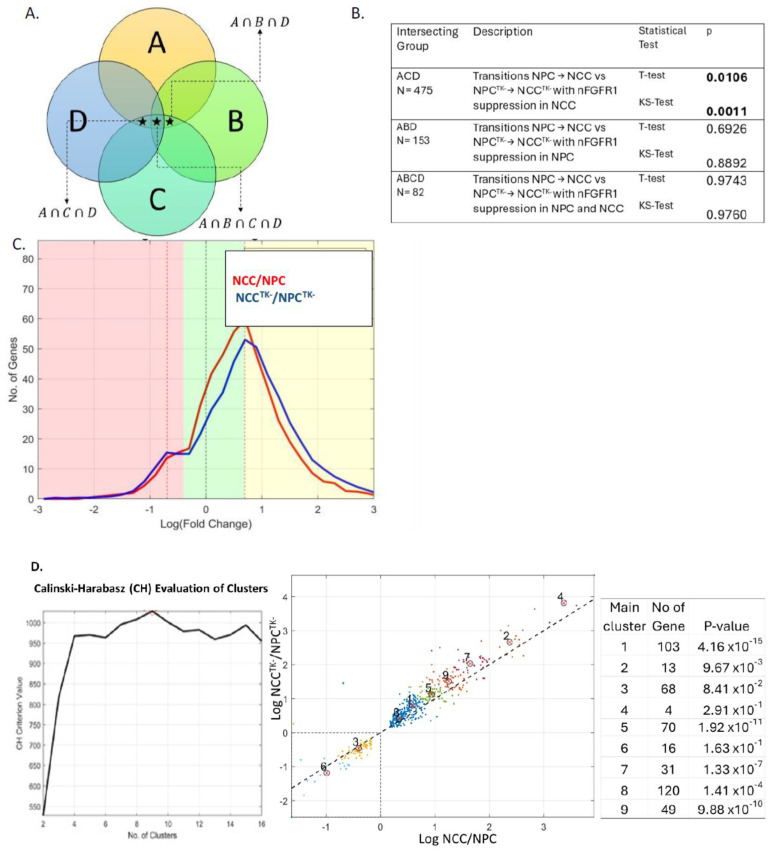
Group comparisons. (**A**) Intersecting groups analyzed. (**B**) Table compares fold gene f activity changes in NCC/NPC vs. NCC^TK−^/NPCT^K−^ in different intersecting subgroups from Figure 1 A. Results of two statistical tests—*t* test and Kolmogorov–Smirnov (KS) test are shown. Only the group A-C-D (nFGFR1 suppression in NCCs) showed a statistically significant difference. (**C**) Histogram for group A-C-D (histograms for A-B-D and A-B-C-D are shown in Appendix A). The logarithmic function redefined a 1-fold change as 0 and a 2-fold change as 1 (shown by the dashed vertical black and red lines) and is divided into three zones: a green zone (genes with low fold changes) and pink and yellow zones (high inhibition and activation fold changes, respectively). (**D**) *k*-means clustering was employed to perform unsupervised data clustering, which partitions *n* observations into *k* clusters, with each observation belonging to the cluster with the nearest mean; *k* values determined by Calinski–Harabasz evaluation identified nine as an optimal number of clusters (left). Note, NCC/NPC inhibitory zones are represented by clusters 3 and 6, and the NCC/NPC activation zone is represented by the remaining clusters; *t* test evaluation of differences between NCC/NPC versus NCC^TK−^/NCC^TK−^ clusters. Distribution of fold changes in the individual clusters is shown in Appendix A.

**Figure 2 ijms-25-05647-f002:**
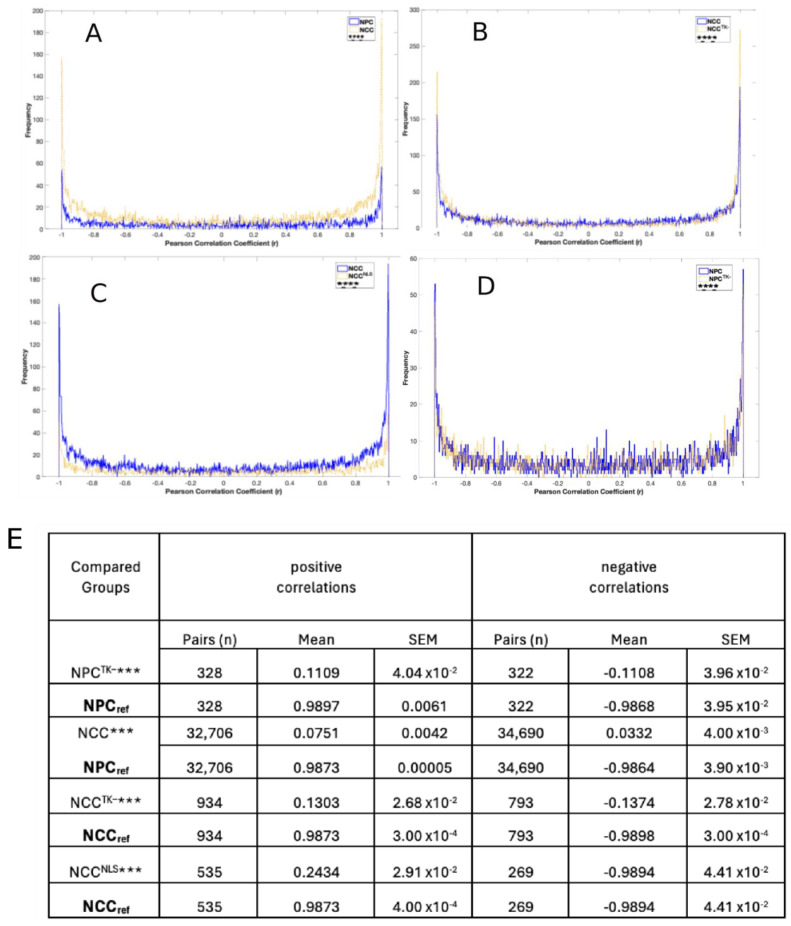
Correlation frequencies. Coordination of nervous system development (NSD) genes: frequency distributions of calculated Pearson coefficients (r). χ^2^ differences between compared conditions for entire range of correlations (−1 to +1): **** *p* < 0.0001); ns, nonsignificant. Gene coordination in NPCs and NCCs (**A**), NCCs and NCC^TK−^s (**B**), NCCs and NCC^NLS^s (**C**), and NPCs and NPC^TK−^s (**D**). From panels (**A**–**D**) we selected the NSD gene pairs which in the reference conditions had r > positive and r < negative critical r threshold values listed in Appendix A. Table (**E**) compares the average Pearson coefficients (*r*_avg_) and the numbers of these strongly correlated gene pairs; *** *p* < 0.001 for both positive and negative correlations. Entropy changes associated with the correlation frequency distributions are listed in Appendix A.

**Figure 3 ijms-25-05647-f003:**
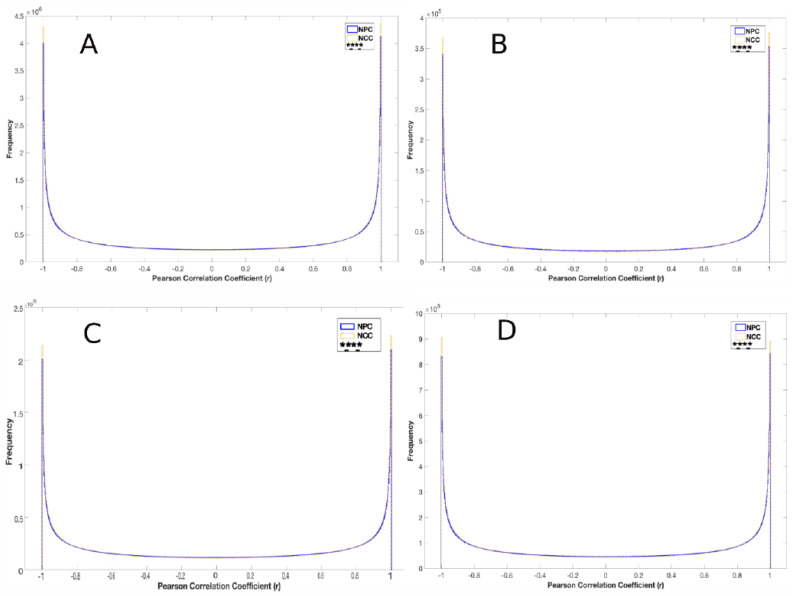
Distributions of gene correlations. χ^2^ tests were used for comparisons of conditions for the entire range (−1 to +1) of Pearson coefficients for NPCs vs. NCCs (**A**–**D**) and for NCCs from patients with schizophrenia versus those from control individuals (**E**). Comparisons of correlations for all expressed genes (16,137 genes) (**A**), for 4646 regulated (Reg) genes (average activity levels changed in NCCs compared to that in NPCs) (**B**), for 11,491 non-regulated (nonReg) genes (average activity levels did not change in NCCs compared to that in NPCs) (**C**), for cross-correlation of Reg and nonReg genes (**D**), 1386 genes commonly dysregulated in schizophrenia patients (Dysreg genes) (**E_1_**), all expressed genes in differentiating NCCs from patients and controls (15,279) (**E_2_**), 13,893 non-dysregulated (nonDysreg) genes (**E_3_**), cross-correlation of Dysreg with nonDysreg genes (**E_4_**). (**F**) Summary of the observed changes in *r* frequency histograms (based on Figure 2A–D, Figure 3A–E, and Appendix A–S6). **** (*p* < 0.0001; NS, not significant) refer to χ^2^ tests of −1 to + 1 range of correlations. (decreases and (+) increases mark direction of changes in strong positive or negative correlations. Entropy changes associated with the correlation frequency distributions are listed in Appendix A.

**Figure 4 ijms-25-05647-f004:**
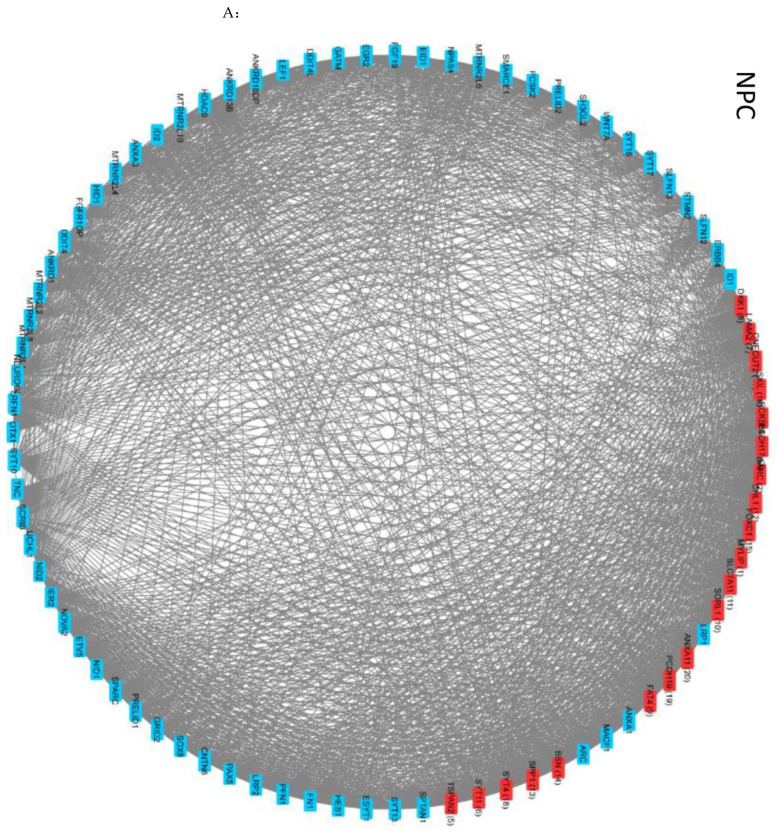
Correlation of the Reg NSD genes: circular network analysis of strongly positively correlated (*r* > 0.99) genes; the strongest 20 correlated genes are marked red. (**A**) Gene activity network (GAN) formed by the 83 strongly correlated genes in NPCs and the GANs formed by those genes in NCCs and in NPC^TK−^s; 83 genes represent Reg genes whose average activity was altered by reduced nFGFR1 function. (**B**) GAN formed by 131 NSD genes strongly positively correlated in NCCs and GANs formed by those genes in NPCs and NCC^TK−^s; 131 genes represent Reg genes whose average activity was altered by reduced nFGFR1 function. (**C**) GAN formed by 90 NSD genes strongly positively correlated in NCCs and the GAN formed by those genes in NCC^NLS^s; 90 genes represent Reg genes whose average activity was altered by overexpression of nFGFR1. (**D**) Circular network characteristics for genes with strong positive correlations; ref, reference condition for a given group of genes.

**Figure 5 ijms-25-05647-f005:**
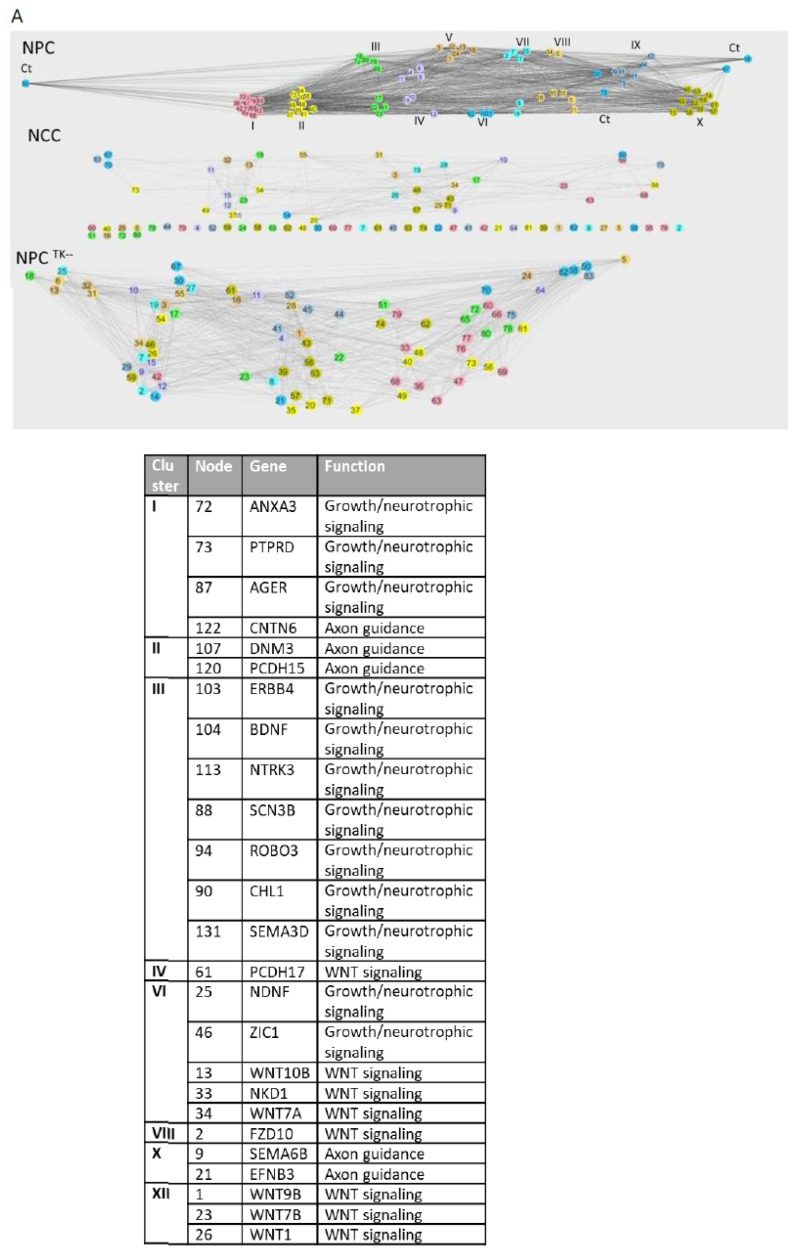
Clustered organization of coordinated GANs. (**A**) Clustered network formed by 83-gene NSD GAN in NPCs and its changes in NCCs and NPC^TK−^s. The NPC network formed nine clusters and eight non-clustered genes referred to as connectors (Cts) labeled with different colors. All genes and cluster assignments are listed in Appendix A. (**B**) Clustered network formed by the 131-gene NSD GAN in NCCs (Figure 3B) and networks of the same genes in NPCs and NCC^TK−^s. The NPC network formed 12 clusters and eight non-clustered connectors (Cts). Genes discussed in the text are listed in table inset. All genes are listed in Appendix A. (**C**) Clustered network formed by 90-gene GAN in NCCs (Figure 3C) and by the same genes in NCC^NLS^s (Figure 3C). The NCC network was composed of four clusters and 14 non-clustered connectors (Cts). All genes are listed in Appendix A.

**Figure 6 ijms-25-05647-f006:**
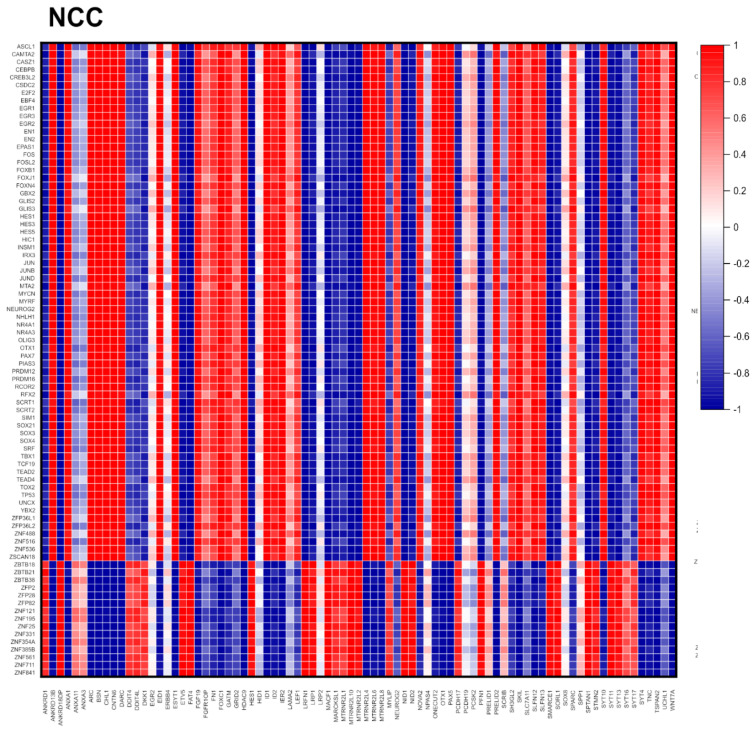
Cross-correlation of transcription factor genes with neurodevelopmental genes. (**A**) Heat maps show cross-correlations of gene pairs among the 4.646 Reg genes. Panels are shown combined in Appendix A. (**B**) Results from quantitative analyses of strongly correlated (red, *r* > +0.96; blue, *r* < −0.96) gene pairs of * *p* < 0.05; *** *p* < 0.001; S, entropy; ΔS, entropy change.

**Figure 7 ijms-25-05647-f007:**
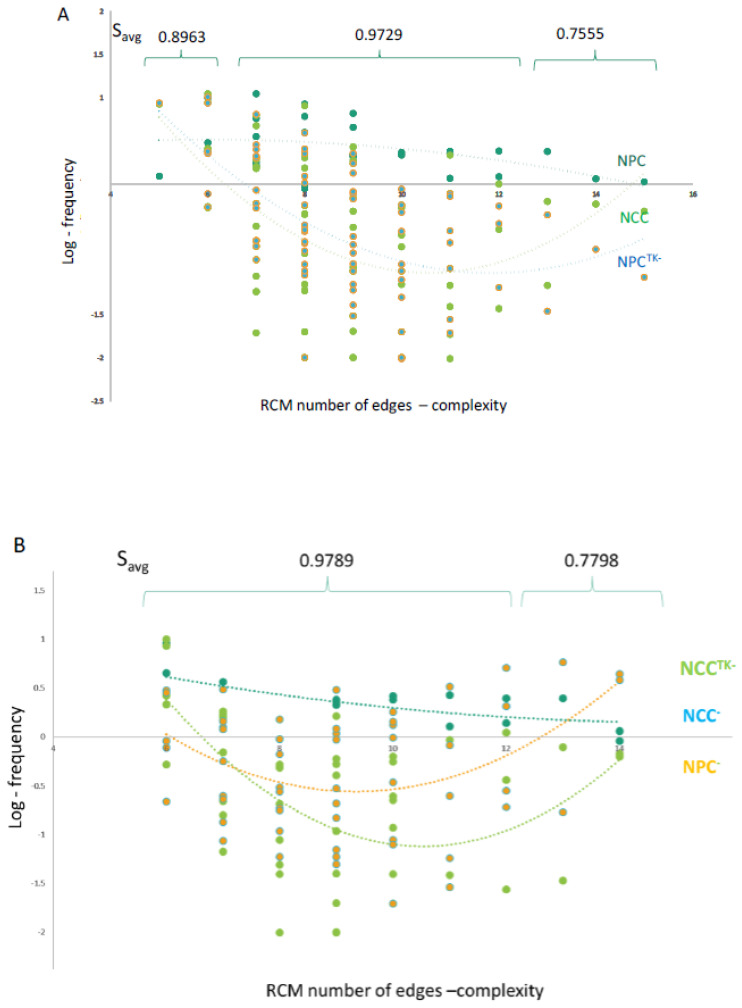
Recurring coordination modules (RCMs). Six-node RCMs overrepresented in GANs formed by the NSD genes: the relationships between complexity (measured by number of edges), entropy, and the frequency of RCMs. (**A**) RCMs of the NPC 83-gene GAN (Figure 3A and Figure 4A) are compared to RCMs formed by the same genes in NPC^TK−^s and NCCs. The average entropy (S_avg_) values were calculated for low-complexity (6 edges), medium-complexity (7–12 edges), and high-complexity (13–15 edges) RCMs. (**B**) RCMs of the NCC 131-gene GAN (Figure 3B and Figure 4B) are compared to the same genes in NCC^TK−^s and NPCs. The S_avg_ values were calculated for medium- and high-complexity RCMs. (**C**) Modeling RCMs as information-transferring electrical circuits. Equivalent electrical circuit for the 6- and 14-node circuits is based on R-L-C circuit theory, which defines a source connected to resistor (R), inductor (L), and capacitor (C). While R, L, and C are connected in series with the source, the output voltage (energy) is measured across C. While the L/C ratio defines oscillations, the R acts as a dampener for the circuit. For the drawn circuit, the more-connected 14-edge network has a lower net inductance, while R and C remain constant. The circuit’s lower L causes more-dampened oscillations. (**D**) The R-L-C response to transient stimulus: the effect of six-node topology on the dampening of the response. The topology of six-node circuits was used to calculate the output response. The less-connected circuits exhibit lower dampening of the circuit activity changes at the onset and offset of the stimulus impulse. The more-connected R-L-C circuit exhibits higher dampening in response to the stimulus. (**E**) Effect of the motif complexity on signal transmission. (**Left**) Two levels of cascading between two identical motifs. (**Right**) Lines show the signal arriving in the second motif and its oscillation (noise) dampening. The six-edge motif generates an under-dampened response, with the information corrupted by noise. The ten-edge motif exerts critical dampening with no signal loss. The 14-edge motif over-dampens the signal—the information is lost.

**Figure 8 ijms-25-05647-f008:**
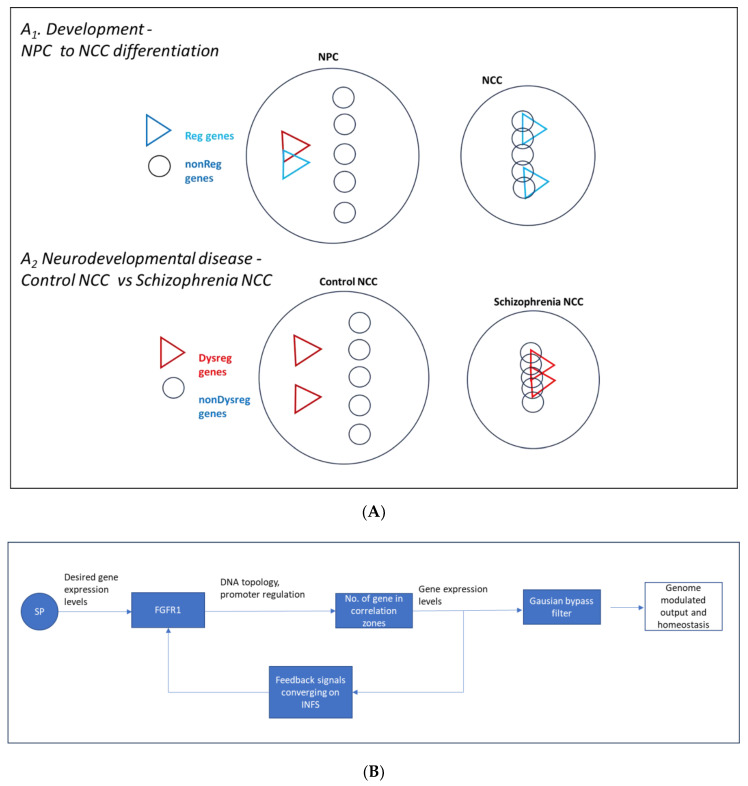
(**A**) Entropy-based genome synchronization model. (**A_1_**): in neural development (NPC → NCC transition), the global expressed genome becomes more synchronized, and the nonReg genes (whose average activity does not change) become more synchronized with each other (lower entropy). The Reg genes (whose average activity changes) become less synchronized to each other but more synchronized to the nonReg genes. The model implies that the nonReg genes increasingly shape the regulation and homeostasis of the Reg genes. This process is controlled by nFGFR1, which makes the NCC Reg genes synchronize less with themselves and more with the nonReg genes. (**A_2_**): in schizophrenia, neuronal development in the brain is impaired and nFGFR1 expression is dysregulated. The population of Dysreg genes (genes with different average activity in NCCs from patients with schizophrenia compared to that in NCCs from control individuals) and the global expressed genome become more synchronized. The nonDysreg genes (whose activity is not different in schizophrenia and control NCCs) also become more synchronized with each other and with the Dysreg genes. The model implies that the nonDysreg genes shape the activity changes of the Dysreg genes and thus brain maldevelopment in schizophrenia. (**B**) nFGFR1 as a proportional-integral-derivative controller and band-pass filter. nFGFR1 maintains gene programming at an ontogenic setpoint by (i) assessing the genome’s output through signals that converge on integrated nuclear FGFR1 signaling (INFS), (ii) determining the difference between the setpoint and the measured output, and (iii) adjusting coordinate gene activities (GANs) through the DNA topology and discriminating fold changes (band-pass filter) such that genome programming closely matches the setpoint.

## Data Availability

The RNA-seq data were deposited in NCBI GEO with accession codes: GSE103307, GSE92874.

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
