# Peer review of "Systems Genome: Coordinated Gene Activity Networks, Recurring Coordination Modules, and Genome Homeostasis in Developing Neurons"

_ijms, 2024, doi:10.3390/ijms25115647_

Round 1
Reviewer 1 Report
Comments and Suggestions for Authors
The present manuscript evaluates the role of nFGFR1 in genome homeostasis. The authors used previously generated (Stachowiak et al., 2017) RNAseq databases.
Q1- The authors used previously generated RNAseq data (Stachowiak et al., 2017). In their previous published research article, they used iPSC lines from 6, 15, and 27-year-old Schizophrenia patients. The effects of nFGFR1 are more prevalent in which age groups?
Q-2 FGFR1 signaling also plays roles in apoptosis and axonal degeneration, indicating its divergent roles. How the author justified deletion of FGFR1 will not affect the neuroinflammation.
Q-3 FGFR1 is highly expressed by glial cells compared to neurons. How only neuronal comparison between the group will provide the proper conclusion of the study.
Q-4 The authors need to discuss the inflammatory aspects of FGFR1 signaling.
Q-5 nFGER1 plays a role in neuronal differentiation. Modulation of nFGFR1 affects which sets of neurons specifically (Inhibitory or Excitatory), which leads to Schizophrenia phenotype.
Q-6 What are the cell type trajectories ratios of excitatory neurons, inhibitory neurons, and glia in cerebral organoids based on preventing nFGFR1-regulated transcription?
Reviewer 2 Report
Comments and Suggestions for Authors
Thank you for submitting the manuscript "Systems genome: coordinated gene activity networks, recurring coordination modules, and genome homeostasis in developing neurons". I found it relevant and interesting to the topic, however, I think that the methods and the quality of presentation must be improved as follows:
1. The methods section contains only minimal info, while the results have a mixture of both results and methods. I would recommend moving the methods info to the correct section and working on clarifying the results.
2. Most of the main figures are unreadable and contain table results that are not well amalgamated with the rest. Please work on better presenting the remain Figures
Detailed info:
1. Figure 1: CD small figures are unreadable and unclear.
Figure 1B: The table is not clear. Can you please refer to the rest of the Figures?
Please add more info in the methods section about the processes undertaken (clustering, approach for testing)
2. Lines 177 - 186: This is clearly part of the methods
3. Figure 2: Unreadable, what is on the axes? Can you refer 2E to the rest of the images? I cannot connect the table to the images; please improve
4. Figure F3: Same as for point 3. - This Figure is unreadable. How is the table connected to the images?
5. Figure 4: Can you please add more info about the colours, strength of the edges, etc.? Include more details about the methods of the network
6. Line 344- Provide more info about the clustering methods
7. Figure 5: Unreadable again. Please improve the presentation and provide info on the methods.
8. How did you assign functions to the genes? Provide info
9. Figure 6. Labels of heatmap are unreadable. I cannot understand what do you want to show
10. Lines 566-595: this is part of the methods
11. Figure 8. 8A I cannot understand the image. The legend used triangles and circles for desreg and reg genes, but the symbols are identical. I am not sure I understand the reasoning behind the shape's location. Could you please create another clearer image? 8B What are the dashed and solid lines? I don't understand what the components are. Can you improve the presentation, please?
Comments on the Quality of English LanguageOverall, the quality of English is sufficient. Minor edits may be necessary
Round 2
Reviewer 1 Report
Comments and Suggestions for Authors
I am satisfied with the response of Dhiman et al..
Reviewer 2 Report
Comments and Suggestions for Authors
I do not have further comments, thank you for adding my suggestions to the manuscript